# Empirical analysis of the text structure of original research articles in medical journals

Nicole Heßler[1], Miriam Rottmann[1¤], Andreas Ziegler[2,3,4]*

**1** Institut für Medizinische Biometrie und Statistik, Universität zu Lübeck, Universitätsklinikum-Schleswig-Holstein, Campus Lübeck, Lübeck, Germany, **2** Medizincampus Davos, Davos, Switzerland, **3** School of Mathematics, Statistics and Computer Science, University of KwaZulu Natal, Pietermaritzburg, South Africa, **4** Department of Cardiology, University Heart & Vascular Center Hamburg, University Medical Center Hamburg-Eppendorf, Hamburg, Germany

¤ Current address: Dr. Hüsing Aktuar GmbH, Bremen, Germany
* Andreas.Ziegler@medizincampus-davos.ch

**Data Availability Statement:** All relevant data are within the manuscript and its Supporting Information files.

**Funding:** The author(s) received no specific funding for this work.

## Abstract

Successful publishing of an article depends on several factors, including the structure of the main text, the so-called introduction, methods, results and discussion structure (IMRAD). The first objective of our work is to provide recent results on the number of paragraphs (pars.) per section used in articles published in major medical journals. Our second objective is the investigation of other structural elements, i.e., number of tables, figures and references and the availability of supplementary material. We analyzed data from randomly selected original articles published in years 2005, 2010 and 2015 from the journals The BMJ, The Journal of the American Medical Association, The Lancet, The New England Journal of Medicine and PLOS Medicine. Per journal and year 30 articles were investigated. Random effect meta-analyses were performed to provide pooled estimates. The effect of time was analyzed by linear mixed models. All articles followed the IMRAD structure. The number of pars. per section increased for all journals over time with 1.08 (95% confidence interval (CI): 0.70–1.46) pars. per every two years. The largest increase was observed for the methods section (0.29 pars. per year; 95% confidence interval (CI): 0.19–0.39). PLOS Medicine had the highest number of pars. The number of tables did not change, but number of figures and references increased slightly. Not only the standard IMRAD structure should be used to increase the likelihood for publication of an article but also the general layout of the target journal. Supplementary material has become standard. If no journal-specific information is available, authors should use 3/10/9/8 pars. for the introduction/methods/results/discussion sections.

## Introduction

Publish and prosper is one of the sayings scientists often encounter. Working in the field of research means constant publishing. The competition among scientists is strong, and journal space is limited. However, the world of publication can be a black box, and writing is challenging for many [1]. Concurrently, the art of scientific communication is rarely taught, and

**Competing interests:** The authors have read the journal's policy and the authors of this manuscript have the following competing interests: Author MR became a paid employee of Dr. Hüsing Aktuar GmbH after the completion of this study. The present position of MR does not alter our adherence to PLOS ONE policies on sharing data and materials. There are no patents, products in development or marketed products to declare. Additionally, AZ is a licensed Tim Albert trainer and has held several courses in the past based on Albert's concept and intends to continue these courses.

scientific writing distinguishes fundamentally from literary writing. Only a few authors focus on the process of writing. One such procedural system was developed by Albert [2], and he demystified article writing using a 10-step process [3]. In one of these steps, Albert's concept looks at a sales model for the article. One key element of this sales approach to publication is that the target audience is the journal editor because the editor is the gatekeeper for acceptance or rejection of an article or even the early rejection, sometimes called desk-rejections. As always the first impression of a manuscript is the best impression [4]. Thrower has provided reasons why he accepted [5] or rejected [6] manuscripts for publication. One important aspect is the presentation of the material [7]. It is clear that authors should follow author instructions of the target journal. The referencing pattern also plays a role, and "a well formatted paper makes the editor happy as he need not to do anything further from his side as far as presentation is concerned" [4]. Reasons for early rejections of articles have been given in an editorial by Froese and Bader [8], and they stated that manuscripts not following the structure a standard article are likely to get rejected. In this context, we need to look at the question what the typical structure of an article is.

The typical structure for the main text of an article in a medical journal is the introduction, methods, results and discussion (IMRAD) structure. Although it was recommended as standard structure at the beginning of the 20th century, it became the primary structure only after 1965, and it was the only article structure in the 1980s [9]. The major change in the mid 1960s is most likely related to a conference of medical editors held at the 19th General Assembly of the World Medical Association [10], during which Hill [11] suggested that research articles should answer the four questions: why did you start, what did you do, what answer did you get and what does it mean anyway. Little research has been done on details of these four sections of an article [2, 12]. Albert [2] an analysis of 50 consecutive articles published after June 1, 1997, from each of the 6 journals Archives of Disease in Childhood (Arch Dis Child), Journal of Pediatrics (J Pediatr), Pediatric Research (Pediatr Res), The BMJ (BMJ), The Lancet (Lancet) and The New England Journal of Medicine (NEJM). He reported the means and standard deviations (SD) of the number of paragraphs per section as given in Table 1. From this, Albert [2] derived the 'typical journal structure', which he calls 2/7/7/6 and which means that 2 paragraphs should be used for the introduction, 7 for methods, 7 for results and 6 for the discussion. Albert [3] was less rigorous and suggested a 2-3/4-6/4-6/5-8 structure. Other recommendations based on Albert's empirical work have also been proposed, such as 2/5/5/4 for shorter articles [13]. Soares de Araújo [12] considered original articles published in the January 2012 and 2013 issues of Arquivos Brasileiros de Cardiologia and the first two issues of the Journal of the American College of Cardiology from the same years and recommended 3/6-9/4-9/

**Table 1. Means and standard deviations for number of paragraphs per section.** Modified from Albert [2], Fig 5.1.

| Journal | Introduction | Methods | Results | Discussion | Structure |
|---|---|---|---|---|---|
| Arch Dis Child | 2.7 ± 1.3 | 6.5 ± 4.0 | 6.1 ± 4.0 | 6.9 ± 2.8 | 3/7/6/7 |
| BMJ | 2.3 ± 0.9 | 6.0 ± 3.7 | 5.9 ± 3.1 | 7.4 ± 2.8 | 3/6/6/7 |
| J Pediatr | 2.6 ± 1.1 | 6.7 ± 3.4 | 7.0 ± 3.9 | 7.3 ± 2.8 | 3/7/7/7 |
| Lancet | 2.6 ± 1.3 | 7.6 ± 3.6 | 6.1 ± 2.9 | 7.0 ± 2.6 | 3/8/6/7 |
| NEJM | 2.6 ± 1.1 | 9.2 ± 3.3 | 8.9 ± 3.8 | 6.9 ± 1.8 | 3/9/9/7 |
| Pediatr Res | 3.0 ± 1.3 | 9.6 ± 3.8 | 6.3 ± 2.9 | 8.5 ± 3.4 | 3/10/6/9 |
| Meta-analysis | 2.6 ± 0.5 | 7.6 ± 1.5 | 6.6 ± 1.4 | 7.2 ± 1.0 | 2-3/6-9/6-7/6-8 |

Arch Dis Child: Archives of Diseases in Childhood, BMJ: The BMJ, J Pediatr: Journal of Pediatrics, NEJM: The New England Journal of Medicine, Pediatr Res: Pediatric Research. Meta-analysis: results from DerSimonian and Laird [14] approach.

≤10 paragraphs. If we meta-analyze the data provided by Albert [2] per section using the Der-Simonian and Laird [14] approach, the recommended structure is 3/8/7/7 (Table 1), i.e., 25 paragraphs in total. Albert's work is, however, more than 20 years old and has not been updated.

The first aim of our article is to provide recent results on characteristics of the structure of original articles published in major medical journals. Since the reporting of studies changed over the past 20 years due to the availability of reporting guidelines, such as the CONSORT [15] or STARD [16] statement, we hypothesize that more recent journal articles have an increasing number of paragraphs over time, especially in the methods section. To this end, we analyze data from randomly selected original articles published in years 2005, 2010 and 2015 from the journals BMJ, The Journal of the American Medical Association (JAMA), Lancet, NEJM and PLOS Medicine (PLOS). We expect that PLOS, an electronic only journal, has more paragraphs than the print journals BMJ, JAMA, Lancet and NEJM because there are, in principle, no page restrictions. The second aim of our work is to expand the statistics to other structural elements, i.e., the number of tables, figures and references as well as the availability of supplementary material. We expect that recently published articles have more likely supplementary material.

## Materials and methods

In original articles published in the English language medical journals BMJ, JAMA, Lancet, NEJM and PLOS in the years 2005, 2010 and 2015, we investigated the number of paragraphs per section. Per journal and year of publication we randomly selected 30 articles, totaling to 450 original research articles. We checked the presence of the IMRAD structure, counted the number of tables, figures and references and checked if supplementary material was available for an article. Data were extracted by M.R.

For each year and each journal, means and SD were calculated for continuous outcomes, and absolute and relative frequencies were reported for the availability of supplementary material. The DerSimonian and Laird [14] approach was used to perform random effect (RE) meta-analyses which allows for variability in the number of paragraphs between journals and over time. These analyses were performed within each journal over the years, within each year over the journals, and for all years and all journals together. Pooled RE estimates and standard errors were calculated. The effect of time was investigated by linear mixed models with journal as RE and year as fixed effect (FE) for the number of paragraphs and the proportion of journals with supplementary information. Effect estimates and corresponding 95% CIs were reported. The specific hypothesis that PLOS has more paragraphs than the print journals was investigated with a mixed linear model with year as RE and journal as FE. Methodological details are provided in the S1 Appendix.

No adjustments were made for multiple testing, and the significance level was set to 0.05 for all analyses. All statistical analyses were done in R version 3.6.2. Data and analysis code in Markdown are available as supplementary material.

## Results

A total of 450 original research articles from 5 medical journals were analyzed. The IMRAD structure was used in all of them. S1 Table displays the descriptive statistics for the number of paragraphs per section for each year and each journal. An increase in the number of paragraphs per section and in total can be seen for all journals over time. The increase is, however, small for JAMA and NEJM. An obvious increase in the methods section can be observed for BMJ and Lancet with an average of 5 and four additional paragraphs, respectively (Fig 1). The

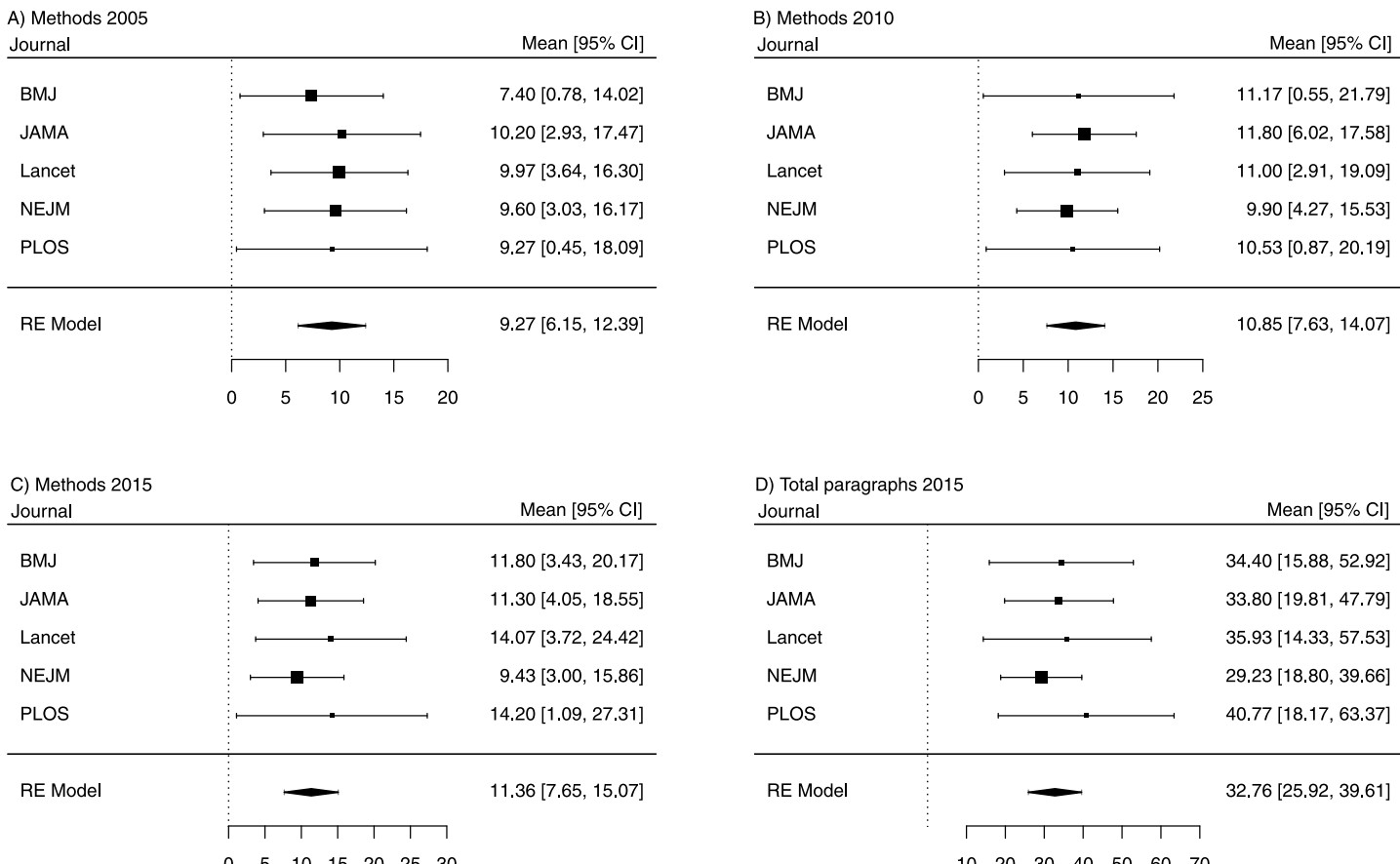

**Fig 1. Forest plots from meta-analyses.** Pooled means of number of paragraphs (black squares) and 95% confidence intervals (CI, lines) are displayed for each journal for A) methods section in 2005, B) methods section in 2010, C) methods section in 2015 and D) the total number of paragraphs in 2015, respectively. Summary statistics (black diamond) was calculated using the random effects DerSimonian & Laird approach [14]. BMJ: The BMJ, JAMA: The Journal of the American Medical Association, NEJM: The New England Journal of Medicine, PLOS: PLOS Medicine.

increase in the number of paragraphs was also visible in PLOS with one, 5, three and two additional paragraphs in the introduction, methods, results and discussion for years 2015 when compared with 2005 (S1 Table and Fig 1).

While the number of figures and tables overall remained rather constant, the number of references increased between 2005 and 2010 and 2015, respectively, with the largest increase in the number of references for BMJ and PLOS (S1 Table). The most striking change is seen for the number of original articles with supplementary material. For example, while no JAMA article had supplementary material in 2005, 97% (29 of 30) had supplementary material in 2015.

Table 2 shows article characteristics by year averaged over the 5 journals. The smallest and largest increase per year for the number of paragraphs was observed for the introduction (0.03 pars. per year, 95% CI: 0.00–0.06) and methods sections (0.29 pars. per year, 95% CI: 0.19–0.39). While the standard article structure was 3/9/9/8 in 2005, it increased to 3/11/10/8 in 2015. On average within 10 years, methods thus increased by two paragraphs and results by one paragraph. This corresponds to an increase of approximately one paragraph every two years (1.08 pars. per 2 years; 95% CI: 0.70–1.45). Consideration of all years and all journals, the pooled standard article structure is 3/10/9/8.

**Table 2. Results of meta-analyses per year over journal.**

| Year | Introduction | Methods | Results | Discussion | Total | Tables | Figures | References | Suppl |
|------|------|------|------|------|------|------|------|------|------|
| 2005 | 3.19 ± 0.53 | 9.27 ± 1.59 | 8.61 ± 1.49 | 7.72 ± 0.94 | 28.55 ± 2.68 | 3.27 ± 0.72 | 1.87 ± 0.62 | 30.65 ± 4.77 | 0.34 ± 0.13 |
| 2010 | 3.08 ± 0.49 | 10.85 ± 1.64 | 8.84 ± 1.70 | 8.19 ± 1.23 | 31.41 ± 3.01 | 3.18 ± 0.66 | 2.45 ± 0.77 | 35.08 ± 5.60 | 0.73 ± 0.09 |
| 2015 | 3.36 ± 0.45 | 11.36 ± 1.89 | 9.61 ± 1.62 | 8.29 ± 1.26 | 32.76 ± 3.49 | 2.90 ± 0.57 | 2.55 ± 0.67 | 34.34 ± 6.25 | 0.97 ± 0.02 |
| All years | 3.22 ± 0.28 | 10.39 ± 0.98 | 9.00 ± 0.92 | 8.00 ± 0.64 | 30.54 ± 1.74 | 3.08 ± 0.37 | 2.26 ± 0.39 | 32.97 ± 3.14 | 0.67 ± 0.11 |
| Change per year | 0.03 | 0.29 | 0.14 | 0.08 | 0.54 | -0.02 | 0.08 | 0.75 | 0.06 |
| 95% CI | 0.00–0.06 | 0.19–0.39 | 0.04–0.23 | 0.01–0.16 | 0.35–0.73 | -0.06–0.02 | 0.03–0.13 | 0.30–1.20 | 0.05–0.07 |
| p-value | 0.050 | < 0.001 | 0.004 | 0.036 | < 0.001 | 0.336 | 0.001 | 0.001 | < 0.001 |

Pooled means and respective standard errors for the number of paragraphs per section, the total number of paragraphs (Total), the number of tables, figures and references from DerSimonian and Laird [14] meta-analysis per year over journals. The proportion of articles with supplementary material (Suppl) is displayed in the last column. The change in the number of paragraphs per year/percent of articles with supplementary material per year from a linear mixed model is displayed together with its 95% confidence interval (95% CI) and p-value.

The number of tables did not alter (-0.02 per year; 95% CI: -0.06–0.02), and figures increased slightly (0.08 per year; 95% CI: 0.03–0.13). Per year 0.75 more references were observed when averaged over the journals (95% CI: 0.30–1.20). This increase is, however, primarily caused by BMJ and PLOS, while the number of references was similar over the years for JAMA, Lancet and NEJM.

During this period, the percentage of articles with supplementary material increased from 39% in 2005 to 93% in 2015. This confirms our hypothesis about the increasing number of supplementary material per years (p < 0.001).

Results per journal pooled over the years are displayed in Table 3 and S1 and S2 Figs. The only considered online only journal PLOS had the largest number of paragraphs in the introduction, results and discussion sections. While PLOS did not have significantly more paragraphs in the methods section, it had approximately one additional paragraph in the introduction and the discussion, respectively. On average, PLOS articles hat 3.52 paragraphs more than the other journals (95% CI: 1.60–5.43). The NEJM had the lowest number of paragraphs per article among the 5 journals considered. It is the only journal still with a 2 before the decimal point for the average number of paragraphs in the introduction. Only BMJ and

**Table 3. Results of meta-analyses per journal over years.**

| Journal | Introduction | Methods | Results | Discussion | Total | Tables | Figures | References |
|------|------|------|------|------|------|------|------|------|
| BMJ | 3.43 ± 0.56 | 9.50 ± 2.38 | 8.78 ± 2.67 | 8.80 ± 1.91 | 28.91 ± 3.75 | 3.36 ± 1.06 | 1.80 ± 0.94 | 31.23 ± 9.10 |
| JAMA | 3.12 ± 0.57 | 11.21 ± 1.96 | 8.94 ± 1.85 | 9.30 ± 1.47 | 32.44 ± 3.49 | 3.48 ± 0.75 | 1.83 ± 0.78 | 35.18 ± 7.33 |
| Lancet | 3.01 ± 0.63 | 11.06 ± 2.29 | 8.79 ± 2.22 | 9.20 ± 2.22 | 31.93 ± 5.20 | 2.98 ± 0.84 | 2.82 ± 0.92 | 32.27 ± 6.89 |
| NEJM | 2.94 ± 0.60 | 9.67 ± 1.82 | 9.08 ± 1.56 | 6.88 ± 0.88 | 28.57 ± 3.02 | 2.66 ± 0.65 | 2.29 ± 0.67 | 30.40 ± 4.87 |
| PLOS | 4.05 ± 0.93 | 10.71 ± 2.98 | 9.45 ± 2.79 | 9.81 ± 2.25 | 34.48 ± 5.62 | 3.38 ± 1.20 | 3.95 ± 1.87 | 46.84 ± 11.58 |
| PLOS vs. others | 1.05 | 0.70 | 0.86 | 0.91 | 3.52 | 0.10 | 1.63 | 14.45 |
| 95% CI | 0.76–1.33 | -0.30–1.69 | -0.09–1.82 | 0.11–0.1.71 | 1.60–5.43 | -0.29–0.49 | 1.14–2.13 | 9.85–19.05 |
| p-value | < 0.001 | 0.200 | 0.080 | 0.030 | < 0.001 | 0.60 | < 0.001 | < 0.001 |

Pooled means and respective standard errors for the number of paragraphs per section, the total number of paragraphs from DerSimonian and Laird [14] meta-analysis per journal over years. The number of additional paragraphs of PLOS Medicine compared with the other four journals as estimated by a linear mixed model is displayed together with the corresponding 95% confidence interval (95% CI) and p-value. BMJ: The BMJ, JAMA: Journal of the American Medical Association, NEJM: The New England Journal of Medicine, PLOS: PLOS Medicine.

NEJM had less than 10 paragraphs on average for the methods, and NEJM had approximately three fewer paragraphs in the discussion compared with the other four medical journals considered. Articles published in NEJM also had the lowest number of tables and references. However, BMJ and Lancet had the lowest number of figures. Unexpectedly, the online only journal PLOS did not have more tables than the print journals (0.10; 95% CI: -0.29–0.49), but 1.63 (95% CI 1.14–2.13) more figures and 14.45 (95% CI: 9.85–19.05) more references compared to the print journals.

## Discussion

All original articles from the 5 major medical journals considered followed the IMRAD structure. From 2005 to 2015, the total number of paragraphs increased by one every two years, and the largest increase was observed for the methods section. While the NEJM had a large number of paragraphs in the Albert analysis from 1996 [2], it had the smallest number of paragraphs averaged over the years. As expected, PLOS had the highest number of paragraphs, and PLOS articles were approximately 3.5 paragraphs longer than articles in the print journals. PLOS also had the largest number of figures and references per article. Compared with 2005, it is now standard for all 5 investigated journals to have supplementary material.

Albert used the number of paragraphs instead of words as measure for the text structure of a scientific article because paragraphs are a more manageable unit than words alone [2]. Paragraphs keep the potential reader interested when they are written so that they "become in effect a series of inverted triangles". This means that the first sentence of a paragraph is the key sentence or topic sentence. The following sentences should be only supportive and elaborate the topic sentence.

In addition, paragraphs are related to items from publication recommendations. The EQUATOR network–EQUATOR stands for Enhancing the QUAlity and Research Transparency Of health Research–provides reporting guidelines and checklists for a wide variety of research and study types as help for authors to make their research transparent. Several journals require authors to fully adhere to these guidelines. Some of the guidelines have been updated over time due to practical experience. For example, the first version of the CONSORT (Consolidated Standard of Reporting Trials) statement was published in 1996 and included 21 items. The last revision, termed CONSORT 2010 provides a checklist with 25 items, of which 12 are further divided into a and b, making 37 items in total. Similarly, the STARD (Standards for Reporting of Diagnostic Accuracy Studies) statement for the reporting of diagnostic studies has also increased in the number of items from its first version, which was published in 2004, to its current edition from 2015. While the first STARD statement has 25 items, the STARD 2015 statement has 30 items, 4 of which have an a/b division, making 34 in total. An increase in the number of paragraphs can therefore be expected if a publication recommendation is updated and the number of items increased.

Based on his empirical work from almost 25 years back in time, Albert recommended the use of 2/7/7/6 paragraphs for the introduction/methods/results/discussion sections. Other authors were less strict in their recommendations [3, 12]. We provide an update about the standard structure for an article, and we now generally recommend 3/10/9/8 paragraphs for the four main sections. The total number of paragraphs thus is approximately 30. This directly leads to a recommendation for the number of words per paragraph. For example, JAMA has a word limit of 3000. With 30 paragraphs this gives approximately 100 words per paragraph in the main text. Soares de Araújo [12] recommended 130 words for cardiovascular journals because these have an upper word count limit of approximately 4000 words after subtraction of references.

Suggestions for the topics of the different paragraphs have been provided by Albert [2] and Soares de Araújo [12]. Their suggestions for topics differ, however, substantially. Specifically, the first paragraph of the discussion is a brief summary of the main findings according to Albert, while the study problem should be discussed again in the first paragraph of the discussion according to Soares de Araújo. In our opinion, the repetition of the study problem is superfluous, and we agree with Albert's general concept of article writing. He suggested 6 topics for the discussion section: 1) summary of main findings, 2) weaknesses of the study, 3) strengths of the study, 4) how it fits in the literature, 5) implications for future research and 6) implications for policy/treatment. If 8 sections are used in the discussion, the summary of the main findings and its fit to the literature are generally expanded by one additional paragraph each, but the topics are not changed per se. Instead, it seems very natural that the fit in the published literature is more elaborated, e.g., by integrating related topics, systematic reviews and judging original studies, such as randomized controlled trials. Writers and scientists also presented concepts how the complex problem of writing a whole journal article can be divided into smaller problems. One such approach was suggested by O'Connor and Holmquist [17], and another concept was developed by Albert [2]. The ideas presented differ substantially, and this is best illustrated from the "summary statements" [17] or the "message" [2]. Albert does not get weary of emphasizing the importance of the message of the paper. The message should have 12 words, should have a verb, should not be a question and, in our experience, should not include an "and". The paper is then streamlined along this single message. This is in contrast to the concept of O'Connor and Holmquist [17] because a manuscript cannot be written according to a single message if there are up to three "conclusions summarizing the major contributions of the manuscript to the scientific community".

In our study, we did not investigate the effect of the article structure on citation frequency. Several studies observed relationships between the length of the title and citation frequency [18–24]. Shorter titles have a higher citation frequency, and the conclusion is: keep the title short. Linguistic complexity of title, abstract and main text has also been studied in relation with citation frequency [20, 25–28]. While no difference was found between citation frequency and linguistic complexity of the main text [28], top ranked journals use a simple language in title and abstract [27]. However, scientific articles are generally difficult to read [29, 30]. For example, the Gunning fog index, which looks at sentence length and word complexity [31], is approximately 17 for scientific articles [32–34]. Although text complexity is reduced are after peer-review, texts are still substantially more complex compared to daily newspaper articles, which have a fog index of 12 [35]. Insurance policies are in contrast even more complex with a fog index of approximately 20 [29].

Authors also investigated the association between citation frequency, page length, the number of references and author recognition [36–38]. It is obvious that larger articles may represent review articles, as may articles with a higher number of references. With these arguments, it can already be expected that articles with more pages and more references have higher citation numbers.

A reviewer of our work has pointed to the importance of investigating the influence of media paying possibly more attention to journal articles in the past years and changes in journal strategies and instructions for authors. These aspects should be assessed in future research.

In conclusion, authors should not only use the standard IMRAD structure to increase the likelihood for publication of their work. They should also take into account the general layout of their target journal. If a journal-specific structure is not at hand, authors should use 3, 10, 9 and 8 paragraphs for the introduction, methods, results and discussion sections, respectively. Supplementary material has become a standard and should be used when deemed appropriate. Authors should be aware that print journals might differ in their structure from online only

journals because of the absence of page limits for online articles. Finally, and most importantly, the instructions to authors of the target journal must definitely be met.

## Supporting information

**S1 File. Data.**
(XLSX)

**S2 File. R Markdown file for analyses.**
(RMD)

**S1 Fig. Forest plots from meta-analyses for introduction, results and discussion section.** Pooled means of number of paragraphs (black squares) and 95% confidence intervals (CI, lines) are displayed for each journal for A) introduction section in 2005, B) introduction section in 2015, c) results section in 2005, D) results section in 2015, E) discussion section in 2005 and F) discussion section in 2015, respectively. Summary statistics (black diamond) was calculated using the random effects DerSimonian & Laird approach (14). BMJ: The BMJ, JAMA: Journal of the American Medical Association, NEJM: The New England Journal of Medicine, PLOS: PLOS Medicine.
(EPS)

**S2 Fig. Forest plots from meta-analyses for tables, figures and references.** Pooled means of number of tables, figures and references (black squares) and 95% confidence intervals (CI, lines) are displayed for each journal for A) tables in 2005, B) tables in 2015, c) figures in 2005, D) figures in 2015, E) references in 2005 and F) references in 2015, respectively. Summary statistics (black diamond) was calculated using the random effects DerSimonian & Laird approach (14). BMJ: The BMJ, JAMA: Journal of the American Medical Association, NEJM: The New England Journal of Medicine, PLOS: PLOS Medicine.
(EPS)

**S1 Table. Descriptive statistics per journal and per year.** Means and standard deviations for the number of paragraphs per section, the total number of paragraphs (Total), the number of tables, figures and references by journal and year of publication. The last column provides absolute frequencies and relative frequencies (in parenthesis) for the availability of supplementary material (Suppl). BMJ: The BMJ, JAMA: The Journal of the American Medical Association, NEJM: The New England Journal of Medicine, PLOS: PLOS Medicine.
(DOCX)

**S1 Appendix.**
(DOCX)

## Author Contributions

**Data curation:** Miriam Rottmann.

**Formal analysis:** Nicole Heßler, Miriam Rottmann, Andreas Ziegler.

**Methodology:** Nicole Heßler, Andreas Ziegler.

**Supervision:** Andreas Ziegler.

**Visualization:** Nicole Heßler, Miriam Rottmann, Andreas Ziegler.

**Writing – original draft:** Nicole Heßler, Andreas Ziegler.

**Writing – review & editing:** Nicole Heßler, Miriam Rottmann, Andreas Ziegler.

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
