## [Decision Letter · Decision Letter 0]

15 Jul 2020

PONE-D-20-15797

Empirical Analysis of the Text Structure of Original Research Articles in Medical Journals

PLOS ONE

Dear Dr. Ziegler,

Thank you for submitting your manuscript to PLOS ONE. After careful consideration, we feel that it has merit but does not fully meet PLOS ONE’s publication criteria as it currently stands. Therefore, we invite you to submit a revised version of the manuscript that addresses the points raised during the review process.

I would like authors to clarify the importance of word limits rather than number of paragraphs or they can clarify why it is important for them in discussion. I would like authors to add discussions on the reasons for changes in publication behavior of authors as well.

We look forward to receiving your revised manuscript.

Kind regards,

Omid Beiki, M.D., Ph.D.

Academic Editor

PLOS ONE

Journal Requirements:

'The author(s) received no specific funding for this work.'

We note that one or more of the authors are employed by a commercial company: Dr. Hüsing Aktuar GmbH

Reviewers' comments:

Reviewer's Responses to Questions

**Comments to the Author**

1. Is the manuscript technically sound, and do the data support the conclusions?

Reviewer #1: Yes

2. Has the statistical analysis been performed appropriately and rigorously? 

Reviewer #1: N/A

3. Have the authors made all data underlying the findings in their manuscript fully available?

Reviewer #1: Yes

4. Is the manuscript presented in an intelligible fashion and written in standard English?

Reviewer #1: Yes

5. Review Comments to the Author

Reviewer #1: Comments of Authors

General Comments

Effective presentation of methods, findings, and importance are of critical importance in obtaining a positive response from journal editors. Important work may not be published if the methods, results, and implications are not stated clearly. My main concern is that the paper is descriptive, but does not deal with why patterns are as they are and why these patterns have evolved. The changes may reflect an increased sophistication of the readership, especially about methods. Large data sets are now more common, and these require more detailed description which reflects both the methods sections and presentation of supplementary material. The increase in supplementary material probably also reflects changes in computer technology, e.g., the cloud. Have the instructions to the authors changed? Has demand for journal space increased, especially among the top-ranked journals? Is the media paying more attention to journal articles than in the past which would affect how papers are written?

Specific Comments

Lines 65 ff. A more compelling justification for focusing on paragraphs is needed. Years ago, I was taught that among the sections, relatively few readers focus on the methods section. This is why this section is sometimes in small print. This section seems to be gaining in importance, but I do not see the link to the number of paragraphs as the best measure of this.

Lines 86-87. The increase in number of paragraphs in the methods section may reflect increasing sophistication of the readership, especially about statistical methods.

Line 106. What is the rationale for using the random effects method? Why is it preferred over the other statistical methods?

Line 120. The paper does not adequately document changes in editorial policy. Have instructions to the authors changed? Changes in word limits and the extent to which they are enforced? This comment also applies elsewhere? It might have been a good idea to have interviewed journal editors to document changes in their policies.

6. PLOS authors have the option to publish the peer review history of their article (what does this mean?). If published, this will include your full peer review and any attached files.

Reviewer #1: No

---

## [Author Response · Author response to Decision Letter 0]

25 Aug 2020

We are very grateful to both reviewers for their feedback and helpful comments which have led to considerable improvement of our manuscript. Reviewer comments are highlighted in blue, and answers to their comments are provided in black in this document. Added and/or altered text is given in italic. These changes are highlighted in the revised manuscript. In addition, we provide a clean version of our manuscript.

Review Academic Editor

I would like authors to clarify the importance of word limits rather than number of paragraphs or they can clarify why it is important for them in discussion. 

We thank the reviewer for the comment and the opportunity to provide an additional justification for the choice of the number of paragraphs as measure in our work. The Introduction, Methods, Results and Discussion (IMRAD) structure can be traced back to Hill (1). His work in the mid 1960 was motivated by inadequate reporting of journal articles. More recently, recommendations for publications have been published for various study designs by the EQUATOR (Enhancing the QUAlity and Research Transparency Of health Research) network (www.equator-network.org). This network does not only provide reporting guidelines but also checklists for a wide variety of research and study types as help for authors to make their research transparent. Practical experience with guidelines has led to revisions. For example, the first version of the CONSORT (Consolidated Standard of Reporting Trials) statement was published in 1996 (2), revised 5 years later (3) and again in 2010, leading to the current CONSORT 2010 statement (4). While the CONSORT statement from 1996 only included 21 items, CONSORT 2010 provides a checklist with 25 items, of which 12 are further divided into a and b, making 37 items in total.

The STARD (Standard for Reporting of Diagnostic Accuracy Studies) statement for the reporting of diagnostic studies has also increased in the number of items. The first statement from 2004 had 25 items, while the revised version from 2015 has 30 items, 4 of which have an a/b division, making 34 in total.

In our opinion, items are closely related to paragraphs, not to the number of words.

Even more importantly, Albert (5) described that authors should look at articles recently published in their target journal and count the number of paragraphs. He stated the number of paragraphs is “a more manageable unit than words alone”. Furthermore, paragraphs keep the potential reader interested when paragraphs are written as a series of inverted triangles (5). The first sentence of a paragraph is the key sentence or topic sentence. The following sentences are only supportive and elaborate the key sentence.

Overall, we are therefore convinced that paragraph is the concept which is simpler to handle, reflects publication recommendations and an important scientific information because of the topic sentences.

We have integrated these arguments into the manuscript, and the added discussion reads:

Albert used the number of paragraphs instead of words as measure for the text structure of a scientific article because paragraphs are a more manageable unit than words alone (2). Paragraphs keep the potential reader interested when they are written so that they “become in effect a series of inverted triangles”. This means that the first sentence of a paragraph is the key sentence or topic sentence. The following sentences should be only supportive and elaborate the topic sentence.

In addition, paragraphs are related to items from publication recommendations. The EQUATOR network – EQUATOR stands for Enhancing the QUAlity and Research Transparency Of health Research – provides reporting guidelines and checklists for a wide variety of research and study types as help for authors to make their research transparent. Several journals require authors to fully adhere to these guidelines. Some of the guidelines have been updated over time due to practical experience. For example, the first version of the CONSORT (Consolidated Standard of Reporting Trials) statement was published in 1996 and included 21 items. The last revision, termed CONSORT 2010 provides a checklist with 25 items, of which 12 are further divided into a and b, making 37 items in total. Similarly, the STARD (Standards for Reporting of Diagnostic Accuracy Studies) statement for the reporting of diagnostic studies has also increased in the number of items from its first version, which was published in 2004, to its current edition from 2015.While the first STARD statement has 25 items, the STARD 2015 statement has 30 items, 4 of which have an a/b division, making 34 in total. An increase in the number of paragraphs can therefore be expected if a publication recommendation is updated and the number of items increased.

I would like authors to add discussions on the reasons for changes in publication behavior of authors as well.

We agree with the reviewer that the publication behavior of authors changed in the last years as well. Thousands of articles are submitted to scientific journals each year. For example, PLOS ONE states on its website (https://journals.plos.org/plosone/s/journal-information, access: August, 10th 2020) that “more than 20,000 new authors are joining the PLOS ONE community every year in over 200 research areas”. Competition among scientists is strong, and there is only limited space for publications. Authors thus become more and more aware of the importance of writing their articles according to the available publication recommendations and instructions of authors to increase their chances of a successful publication.

We have altered the introduction which now reads:

Publish and prosper is one of the sayings scientists often encounter. Working in the field of research means constant publishing. The competition among scientists is strong, and journal space is limited. However, the world of publication can be a black box, and writing is challenging for many (1). Concurrently, the art of scientific communication is rarely taught, and scientific writing distinguishes fundamentally from literary writing. Only a few authors focus on the process of writing. […]

Reviewer 1

First of all, we want to thank Reviewer 1 for the useful suggestions. We fully agree that several suggestions, such as the influence of media, change of the instructions for authors and interviewing editors are of importance. However, these suggestions are beyond the scope of our own work, and they should be addressed in future research. We added the suggestions in the discussion, and we provide further information below.

General Comments

Effective presentation of methods, findings, and importance are of critical importance in obtaining a positive response from journal editors. Important work may not be published if the methods, results, and implications are not stated clearly. My main concern is that the paper is descriptive, but does not deal with why patterns are as they are and why these patterns have evolved. The changes may reflect an increased sophistication of the readership, especially about methods. Large data sets are now more common, and these require more detailed description which reflects both the methods sections and presentation of supplementary material. 

In our reply to the first comment from the Academic Editor, we have hopefully addressed this aspect in sufficient detail. 

Furthermore, we focus again to the CONSORT statement: CONSORT 2010 (4) provides a minimum set of recommended items for reporting a randomized clinical trial which improves the quality of adequate reporting of a trial. Due to that, readers may better understand specific aspects of the trial, such as its design, conduct, analysis or interpretation of the results. Even more, readers may assess the validity of the trial results. Publication recommendations thus facilitate complete and transparent reporting. In our opinion, this increases the chance of a publication in the target journal or may even be a prerequisite. In 1996, CONSORT was a 21-item checklist (2). The revised CONSORT 2001 statement included a 22-item checklist (3). In greater detail, items that were previously combined were now separated, e.g., outcomes (item 6) and sample size (item 7) were separated. Moreover, some items requested additional information.

In 2010, three more items were added. This reflects the need of more information provided by authors. 

In summary, reporting guidelines use items to structure the scientific work, and in our opinion items reflect paragraphs.

We have added these arguments to the discussion which now reads:

Albert used the number of paragraphs instead of words as measure for the text structure of a scientific article because paragraphs are a more manageable unit than words alone (2). Paragraphs keep the potential reader interested when they are written so that they “become in effect a series of inverted triangles”. This means that the first sentence of a paragraph is the key sentence or topic sentence. The following sentences should be only supportive and elaborate the topic sentence.

In addition, paragraphs are related to items from publication recommendations. The EQUATOR network – EQUATOR stands for Enhancing the QUAlity and Research Transparency Of health Research – provides reporting guidelines and checklists for a wide variety of research and study types as help for authors to make their research transparent. Several journals require authors to fully adhere to these guidelines. Some of the guidelines have been updated over time due to practical experience. For example, the first version of the CONSORT (Consolidated Standard of Reporting Trials) statement was published in 1996 and included 21 items. The last revision, termed CONSORT 2010 provides a checklist with 25 items, of which 12 are further divided into a and b, making 37 items in total. Similarly, the STARD (Standards for Reporting of Diagnostic Accuracy Studies) statement for the reporting of diagnostic studies has also increased in the number of items from its first version, which was published in 2004, to its current edition from 2015.While the first STARD statement has 25 items, the STARD 2015 statement has 30 items, 4 of which have an a/b division, making 34 in total. An increase in the number of paragraphs can therefore be expected if a publication recommendation is updated and the number of items increased.

The increase in supplementary material probably also reflects changes in computer technology, e.g., the cloud. 

It could well be that we have misunderstood the reviewer, and we partly agree with the reviewer. In our opinion, there are two main reasons for the increase of supplementary material. First, the research itself and the reporting of research is getting more complex. Study data sets are getting larger, and the increased data sets need more detailed and longer descriptions, especially in the Methods and the Results sections. Results from primary or main analyses are generally presented, while sensitivity analyses and additional findings are in the supplement. This allows to the keep focus on the main aspects and findings of the article. Second, and this might be the argument of the reviewer, electronic publishing permits longer articles because, in principle, there are no page restrictions. The possibility of electronic publications (pdf articles in addition to printed articles) provides the space to place further information in supplementary material.

Have the instructions to the authors changed? Has demand for journal space increased, especially among the top-ranked journals? Is the media paying more attention to journal articles than in the past which would affect how papers are written?

We fully agree with the reviewer that this aspect is of interest. However, it is beyond the scope of our work and should be left for future research. We have added this point in the discussion of our revised manuscript.

The demand for journal space has substantially increased. This is well reflected by the number of journals which increased over time. 

For example, the number of journals listed in 

Index Medicus and MEDLINE (https://www.nlm.nih.gov/bsd/index_stats_comp.html) increased substantially over time. Furthermore, there are more authors per year (see, e.g., for the USA from 2011 to 2019: https://www.statista.com/statistics/572476/number-writers-authors-usa/). Space in high ranked journals has not increased accordingly.

Media might be important as well. However, the role of media is beyond the scope of our work. We have added this aspect in the discussion, and the text now reads:

A reviewer of our work has pointed to the importance of investigating the influence of media paying possibly more attention to journal articles in the past years and changes in journal strategies and instructions for authors. These aspects should be assessed in future research.

\f

Specific Comments

Lines 65 ff. A more compelling justification for focusing on paragraphs is needed. Years ago, I was taught that among the sections, relatively few readers focus on the methods section. This is why this section is sometimes in small print. This section seems to be gaining in importance, but I do not see the link to the number of paragraphs as the best measure of this.

Above, we have argued that the number of paragraphs is naturally related to the publication recommendations and revised our text in the discussion.

Lines 86-87. The increase in number of paragraphs in the methods section may reflect increasing sophistication of the readership, especially about statistical methods.

In our opinion, the increase of the number of paragraphs, particularly in the method section, is reflected by the available publication recommendations. We have incorporated the arguments in the Discussion.

Line 106. What is the rationale for using the random effects method? Why is it preferred over the other statistical methods?

We chose the random effect (RE) model because this meta-analysis model permits the true effect sizes, i.e., the number of paragraphs of a section, to vary between the journals and the years. This heterogeneity is taken into account in a meta-analytical model with RE. In contrast, the meta-analytical model with fixed effects (FE) assumes that the number of paragraphs of a section is always the same for all journals and at all time points. Only random fluctuation would be permitted in this model. However, this assumption does not correspond with our assumptions that 1. the number of paragraphs of a section varies between the considered medical journals and 2. the number of paragraphs of a section varies over time. 

We have rephrased the methods as follows:

The DerSimonian and Laird (14) approach was used to perform random effect (RE) meta-analyses which allows for variability in the number of paragraphs between journals and over time. 

Line 120. The paper does not adequately document changes in editorial policy. Have instructions to the authors changed? Changes in word limits and the extent to which they are enforced? This comment also applies elsewhere? It might have been a good idea to have interviewed journal editors to document changes in their policies.

We fully agree with the reviewer that changes in the editorial policy might have played a role. However, the change of the instructions of authors is beyond the scope of our work.

We have rewritten the discussion which now reads:

A reviewer of our work has pointed to the importance of investigating the influence of media paying possibly more attention to journal articles in the past years and changes in journal strategies and instructions for authors. These aspects should be assessed in future research.

References

(1) Hill AB. The reasons for writing. BMJ. 1965; 2:870.

(2) Begg C et al. Improving the quality of reporting of randomized controlled trials. The CONSORT statement. JAMA. 1996; 276:637-9

(3) Moher D, Schulz KF, Altman D; CONSORT Group (Consolidated Standards of Reporting Trials). The CONSORT statement: revised recommendations for improving the quality of reports of parallel-group randomized trials. JAMA. 2001; 285:1987-91

(4) Schulz KF, Altman DG, Moher D, CONSORT Group. CONSORT 2010 Statement: Updated Guidelines for Reporting Parallel Group Randomised Trials. PLOS Med 2010;7: e1000251.

(5) Albert T. Winning the Publications Game: The Smart Way to Write Your Paper and Get It Published. 4th ed. Boca Raton: CRC Press; 2016.

---

## [Editor Report · Decision Letter 1]

24 Sep 2020

Empirical Analysis of the Text Structure of Original Research Articles in Medical Journals

PONE-D-20-15797R1

Dear Dr. Ziegler,

We’re pleased to inform you that your manuscript has been judged scientifically suitable for publication and will be formally accepted for publication once it meets all outstanding technical requirements.

Kind regards,

Omid Beiki, M.D., Ph.D.

Academic Editor

PLOS ONE
---

## [Editor Report · Acceptance letter]

28 Sep 2020

PONE-D-20-15797R1 

Empirical Analysis of the Text Structure of Original Research Articles in Medical Journals 

Dear Dr. Ziegler:

I'm pleased to inform you that your manuscript has been deemed suitable for publication in PLOS ONE. Congratulations! Your manuscript is now with our production department. 

Kind regards, 

on behalf of

Dr. Omid Beiki 

Academic Editor

PLOS ONE